# Pilot Study on the Evaluation of the Diet of a Mexican Population of Adolescents

**DOI:** 10.3390/pediatric17040078

**Published:** 2025-07-22

**Authors:** Karen Rubí Escamilla-Gutiérrez, Alejandra López-García, Nelly del Socorro Cruz-Cansino, José Alberto Ariza-Ortega, Eli Mireya Sandoval-Gallegos, Esther Ramírez-Moreno, José Arias-Rico

**Affiliations:** 1Academic Area of Nutrition, Institute of Health Sciences, Autonomous University of the State of Hidalgo, Pachuca 42130, Mexico; kescamillagtz@gmail.com (K.R.E.-G.); ncruz@uaeh.edu.mx (N.d.S.C.-C.); jose_ariza@uaeh.edu.mx (J.A.A.-O.); eli_sandoval7987@uaeh.edu.mx (E.M.S.-G.); 2Academic Area of Nursing, Institute of Health Sciences, Autonomous University of the State of Hidalgo, Pachuca 42130, Mexico; lo294624@uaeh.edu.mx

**Keywords:** adolescent, diet, eating, antioxidants

## Abstract

**Background:** Adolescence is characterized by physical and psychosocial changes. This implies modifying or implementing correct nutritional habits at an early age, which would have an impact on a healthy adult life. **Objectives:** The FFQ of dietary patterns has not been estimated in the population of adolescents. Therefore, conducting a pilot, cross-sectional, comparative, and correlational study, we sought to identify frequently consumed foods in an adolescent population. **Methods:** As part of the methodology, a Food Frequency Questionnaire (FFQ) was given to a non-probability convenience sample of 178 subjects aged 14 to 19 years to evaluate the most frequently consumed foods. **Results:** This study showed that the diet of Mexican adolescents was characterized with little variability in the foods consumed: 28.20% of the population had a good consumption of milk (1 to 5 serving/day), 16.50% of sugars, and 16% of cereals, while only less than 15% had a good consumption of source animal foods, fruits and vegetables, oils, and fat. **Conclusions:** Mexican adolescents have demonstrated that their diet is poorly varied. Adolescence is an important period in life that can define habitual dietary intake, and therefore, it is crucial to promote healthy eating at this age. Further research and appropriate public policies are needed.

## 1. Introduction

Adolescence is the period of life between ages 10 and 19, characterized by significant physical and psychological changes [1]. It is a critical stage for establishing habitual food intake that will remain for the rest of life and is even considered as the last chance to prepare people to have a healthier life [2]. However, it is important to remember that eating habits and dietary patterns acquired during early childhood are likely to persist into adulthood. Therefore, it is essential to continuously implement strategies in the school, home, etc., environments that help preserve them.

The American Heart Association (AHA) has suggested that diets, from early childhood and adolescence, should be based on the consumption of fruits and vegetables, whole grains, low-fat and fat-free dairy products, beans, fish, and lean meats. They emphasize that foods with a reduction in saturated and trans fats, cholesterol, and added sugar and salt allow the maintenance of a healthy state through the adequate intake of macro- and micronutrients and bioactive compounds such as the antioxidants from diverse food sources.

However, fruit and vegetable consumption among adolescents is often insufficient [3,4,5]. These compounds play an important role during growth and development. In this age group, metabolic demands are greater, making them more susceptible to not achieving adequate intake, hence, the importance of assessing their consumption and developing strategies to promote their inclusion in the regular diet. The Mexican National Health and Nutrition Survey (ENSANUT 2018-Encuesta Nacional de Nutrición y Salud) [6] classifies foods into recommended for daily consumption (fruits, vegetables, legumes, water, eggs, unprocessed meats, dairy products, nuts, and seeds) or not recommended (processed meats, fast food and fried Mexican snacks, snacks, sweets and desserts, sweet cereals, and sweetened beverages).

The dietary intake of adolescents includes processed foods that are high in fat and sugars, which are associated with overweight and obesity [4]. In ENSANUT 2018, only 35.2% of adolescents consume fruits, and only 24.9% include vegetables in their daily diet [6]. The objective of this study is to estimate the frequency and consumption of foods in a sample of Mexican adolescents (men and women) who attend the school.

## 2. Materials and Methods

### 2.1. Subjects and Study Design

A pilot, cross-sectional, comparative, and correlational study was conducted on a non-probabilistic sample of 14- to 19-year-old (15.94 ± 1.00 years) adolescents recruited from a high school located in Mineral de la Reforma, Hidalgo, Mexico, in July 2018. The sample size was based on convenience. A total of 178 subjects were included, with 48.9% being women and 51.1% being men. The study was approved by the Ethics and Research Committee of the Instituto de Ciencias de la Salud (ICSa) of the Universidad Autónoma del Estado de Hidalgo (UAEH), in México (code CEEI-041-2019). All the participants and their parents or legal tutors signed an informed consent to participate in the study. The inclusion criteria were (1) students enrolled in high school, (2) subjects between 14 and 19 years of age, (3) adolescents in their first through fifth semesters of high school, and (4) students who submitted informed consent forms signed by themselves and their parents or tutor. Exclusion criteria: (1) Adolescents who are pregnant. (2) Adolescents with a gastrointestinal disease.

### 2.2. Food Intake

In this study, data were collected using the Food Frequency Questionnaire (FFQ), a validated semi-quantitative instrument for Mexican adolescents [7]. The FFQ measures food frequency over the past two weeks and consists of three sections. The first contains questions related to subject identification and socioeconomic data; the second contains a list of 99 foods from the Mexican Equivalent Foods System (MEFS); and the final section collected information on food preparations [8]. In Mexico the MEFS is a tool that establishes food equivalents corresponding with a portion or ration of food whose nutritional contribution is similar to those of the same group in quality and quantity, allowing them to be interchangeable with each other and facilitating communication between health professionals and their patients, which includes Mexican foods in the following groups: (1) vegetables; (2) fruits; (3) cereals and tubers; (4) legumes; (5) animal origin foods; (6) milk; (7) oils and fats; and (8) sugars. The FFQ was evaluated and its reproducibility and internal consistency validated by means of a Cronbach’s Alpha analysis, obtaining a very similar value in both applications (FFQ1 α = 0.91; FFQ2 α = 0.92). Test–retest reproducibility was confirmed by the Wilcoxon test, and no significant differences were found between both applications (Z = ≤ 1.96; *p* > 0.05). Therefore, it could be a valid and reliable instrument to measure food intake of adolescents. In addition, the Intraclass Correlation Coefficients (ICC) showed moderate correlation in calcium (0.28) and slight in energy (0.10), protein (0.12), and lipids (0.12). And the results were confirmed by means of the Bland–Altman method [7].

The questionnaire (Appendix A) was carried out once and included consumption frequencies of 4–5 per day, 2–3 per day, 1 per day, 5–6 per week, 2–4 per week, 1 per week, 2–3 per fortnight, and never.

The time frame evaluated by the FFQ corresponded to the last two weeks from the day of the survey. The final section was a table asking participants to indicate their most frequent food preparation methods (Appendix B). This instrument was administered in groups with clear instructions. The adolescents answered the questionnaire manually based on their consumption, and with the support of food models (Figure 1) to facilitate understanding of serving sizes. In addition, home-made portion sizes: teaspoon (5 mL), spoon (15 mL), and cup (240 mL) were presented to adolescents according to the sizing proposed by the MEFS. Each one was guided by one of our trained researchers. The calculation was as follows: Of the eight frequency categories included in the FFQ, the sum of the first three categories (4–5/day, 2–3/day, and 1/day) was used to obtain the consumption of foods in each group. The groups of food most consumed were those with a percentage equal to or greater than 15% of the adolescent population. The amount in grams of each food consumed daily by adolescents was calculated by analyzing data provided by the Food Consumption Questionnaire. Daily intake was calculated based on the total grams obtained from the FFQ and using the MEFS energy reference for each food.

In addition, information was collected in a table of the foods they consumed most frequently, with a profile of reported antioxidant compounds.

### 2.3. Statistical Analysis

The results were expressed as mean ± standard deviation. The data entry and statistical analysis were conducted using SPSS (Statistical Package for Social Sciences), version 24 for Windows. A descriptive analysis was performed for the Food Frequency Questionnaire (FFQ), *t*-tests were used for comparisons with a significance level of <0.05, while the Kolmogorov–Smirnov test was employed to determine normality (*n* > 50).

## 3. Results

### Food Intake

According to the food intake measured through of a semi-quantitative Food Frequency Questionnaire (FFQ), whole milk was the most frequently consumed food among adolescents, since 64.3% expressed having consumed milk from 1 to 5 servings per day, followed by corn tortilla and table sugar, which stood out for being consumed for almost half of the population (47.3% and 42.2%, respectively) (Figure 2). For the fruit group, 30.6% of the subjects reported having consumed apples, and among foods of animal origin, chicken and turkey ham were the most frequently consumed (23.6%), while the vegetable group was husk tomato with 23.1%, in the legumes group were beans (20.6%), and finally, avocados and whole cream accounted for approximately 15% of consumption in the oils and fats category.

Figure 3 represents the mean of the frequencies of each individual food within each food group [8]. It includes prepared food derived from the Food Frequency Questionnaire (pizza, paste, hamburger, and chilaquiles), showing the percentage of adolescents who consumed one to five portions per day of food groups. These data corresponded to the first three categories (vegetables, fruits, and cereals) of frequencies included in the FFQ that are considered foods of high consumption. In general, only 8.6% to 28.2% of adolescents consumed one to five daily servings from each of the food groups. These results show which foods were most widely consumed in the diet of the adolescent group. Dairy products accounted for the highest percentage of the population (28.2%), followed by sugars (16.5%) and cereals (16%). The fruits and animal source foods were close to the highest consumption percentage (14.80 and 14.70%, respectively) in the adolescent sample. And with lower consumption were vegetables, oils and fats, and legumes (11.20%).

As illustrated in Table 1, the foods consumed most frequently on a daily basis, in terms of edible portions per person, and their equivalent values, as determined by the FFQ, are mostly husk tomato, apple, melon, pasta, beans, and avocado. The results obtained in the present study show that the average consumption of fruits by the adolescents was 382.59 g/day and vegetables 269.93 g/day. Considering the Mexican Equivalent Foods System (MEFS) [8], the consumption of vegetables (2.90), legumes (0.85), and sugars (0.29) does not cover these recommendations according to the amount of food evaluated in this study, while the fruits (3.70), cereals (8.58), and oils and fats (7.59) revealed that these food groups exhibited higher consumption, covering the established recommendations for food consumption by equivalents.

## 4. Discussion

According to the FFQ data, the diet of this group of Mexican adolescents was characterized by the consumption of whole milk, corn tortilla, and table sugar. The consumption of fruit and vegetables does not meet national recommendations on a quantitative level. This is due to a low level of dietary diversity, which also leads to a low consumption of fiber and bioactive compounds, such as antioxidants. According to NOM-043-SSA2-2012 [10], which promotes health and education regarding food, stipulates that at least half of the content of one plate should consist of fruits and vegetables, with seasonal products being preferred. The remaining should consist of grains and cereals (22%), legumes (15%), foods of animal origin (8%), and 5% of natural oils and fats.

According to this study, only sugars, milk (dairy products), and cereals (as tortilla) were considered foods of frequent consumption in the adolescent sample. The fruits and animal source foods were close to the highest consumption percentage, and with lower consumption were vegetables, oils and fats, and legumes. The findings underscore the need for interventions aimed at diversifying and improving their diets. These results are close to those found in other national studies in the same population. The ENSANUT 2020–2022 reports that adolescents in rural areas of Mexico maintained a high consumption of animal origin foods, sugars, dairy products, and fruits (46.5, 44, 43.5, and 39.2%, respectively), while their vegetable and legume consumptions were 22.1% and 27.6%, respectively [11]. This is probably due to the direct availability of local foods in rural areas. In urban areas, the consumption rates were as follows, with higher intake of unprocessed meats, sugars, and dairy products, 59.5, 47.2, and 46.8%. Fruits (39.1%), vegetables (31.4%), and legumes (20.2%) approached lower consumption percentages [11]. Therefore, the frequency intake of milk, sugars, cereals, and legumes in our study was lower than the reported results for different states of Mexico in cereals [5,12], milk, and sugars [5].

In the international context, the consumption recommendations of fruits and vegetables, the World Health Organization (WHO) suggests a daily consumption of fruits and vegetables of at least 400 g/day to have a healthy diet, while the Spanish Society of Nutrition Community recommends 300 to 400 g/day of vegetables and 360 to 600 g of fruits per day [9]. The results obtained in the present study show that the consumption of fruits (382.59 g/day) and vegetables (269.93 g/day) reported by a percentage (11–14%) of adolescents was close to both recommendations, although with little variability in vegetable foods.

According to Mexican recommendations for food equivalents in an adequate diet (4 equivalents for vegetables, 3 for fruits, 10 for cereals, 2 for legumes, 5 for oils and fats, and 3 for sugars), fruit, oils, and fat consumptions were adequate, while vegetables, cereals, legumes, and sugar consumptions were low. In the case of sugars, consumption was low according to the equivalents consumed and foods evaluated (chocolate powder), since one equivalent is one tablespoon of the product [8]. However, Mexico generally has a high sugar consumption rate, as adolescents are frequently exposed to ultra-processed foods containing added sugars or prepared products (sweets, desserts, breads, preserves, and beverages). Consequently, numerous public policies have been implemented to regulate consumption. These include restrictions on the availability of prepared and processed foods and beverages in elementary schools [13] and a tax on non-staple, high-calorie foods and sugary beverages [14].

International recommendations by the Academy of Nutrition and Dietetics [15] is 4–5 servings of fruits and vegetables, 6–8 servings of cereals, and 2–3 servings of oils and fats/day, while the recommendation of legumes and sugars are per week (4–5 servings and 5 servings, respectively), which have variability with what is established as recommendations for Mexico. On the other hand the proportion of adolescents (14.7 and 11.2%) who consumed one to five portions per day of vegetable and fruit groups in this study was lower than the result reported for adolescents from Mexico City, Michoacan, and Guadalajara states which was 26% [4,12] and lower than that reported by ENSANUT, INEGI, and INSP 2018 [6] (35.2% and 24.9%) in fruits and vegetables, respectively.

The consumption of fruits and vegetables by Mexican adolescents is not always sufficient [3,4,5], and thus, the intake of bioactive compounds is low. Thus, knowing the quantity of foods frequently consumed is relevant to determining the future impact these foods will have on short and long-term health status. The role of these compounds during adolescence is important due to the functions they play during growth and development. In this age group, metabolic demands are greater, making them more susceptible to not achieving adequate intake, hence, the importance of assessing their consumption and developing strategies to promote their inclusion in the regular diet.

The nuts and seeds present the lowest consumption in adolescents of rural and urban areas (≈ 2%). Nuts are not included in the Mexican food intake recommendations, but the Spanish Society of Community Nutrition suggests an intake of 20 to 30 g/day [9]. Mexico does not have a high consumption of these products due to a lack of habit and perhaps high cost, in contrast to other European countries where nuts are a characteristic food, particularly of the Mediterranean diet [16]. However, in our study, the nut intake was relevant, such as Japanese peanuts (processed product), natural peanuts, and pecan nuts with 17.21 g per day.

Moreover, in our study, the intake of legumes was lower compared with previous reports [3,12], probably because our subjects came from an urban setting where the intake of legumes is often lower [3]. In general, bean consumption has decreased due to changes in consumer preferences favoring processed foods [17]. In other countries (Peru, Honduras, El Salvador, and Spain), intake of different food groups by adolescents is higher than our results, but still insufficient to cover their nutritional recommendations [18,19,20,21].

It is common for adolescents to skip meals; include more snacks and fast food in their diet [22,23]; and consume less fruits, vegetables, and cereals [23]. In addition, adolescents tend to have body image concerns that can lead to the adoption of restrictive diets [20]. Although they may want to carry a natural and healthy diet, they may have a mistaken perception about healthy food, lack of knowledge about healthy eating habits, and the impact of making wrong nutritional choices [24]. In addition, the accessibility to industrialized food and the influence of communication media promote unbalanced diets [24,25]. Because of this, new ways of implementing public policies on junk food have been created in Mexico. Regulatory proposals have promoted the regulation of these products and government advertising, such as the “*plate of good eating*” and “*The Drinkwell Jar*”; guidelines for the sale of junk food in schools; and taxes on high-calorie products [26]. In addition, taking into account the obesogenic environment in Mexican schools, the modification of NOM-051-SCFI/SSA1-2010 [27] regarding general labeling specifications for prepackaged foods and non-alcoholic beverages, including commercial and health information, was promoted where the main objective was to identify commercial and health requirements in a practical way, using precautionary seals on processed product labels when there are excess in calories and critical nutrients (fats, sodium, and sugars) [27,28].

Mexico has a large variability in fruits and vegetables and whole grains [29], but their consumption has decreased due to an income growth by refined carbohydrates, higher content of saturated fat, and processed foods [30,31]. The consumption of food is influenced by a variety of social, economic, and cultural characteristics of a particular region or population [10]. Different studies show that adolescents lack an adequate consumption of food groups [4,5,12,18,19,20], which is worrying considering that insufficient fruit and vegetable intake is one of the main ten mortality risk factors worldwide [32], and is linked to many diseases, such as gastric cancer [20,33,34], ischemic heart diseases, and cerebrovascular accidents [20,35,36,37].

On the other hand, Mexicans typically eat as a family, and these are determined by various factors, such as attitudes, knowledge, beliefs, and affordability, which in turn could determine family member access to foods [38]. In Mexico, lower socioeconomic factors favor the affordability of fresh foods and greater availability of processed foods, or social factors and greater availability of processed foods, or social factors. A further point to be considered is that soda is widely regarded as an unhealthy product. However, a significant proportion of sweetened beverages, such as juices, are regarded as healthy by children, parents, and teachers [38].

During adolescence, health status is influenced by several factors, among which dietary patterns are a crucial element of lifestyle in terms of preventing and treating metabolic and chronic diseases, so it is important that the diet of adolescents is varied and has a combination of foods that are rich in antioxidant and anti-inflammatory nutrients [39]. The Mediterranean diet has been considered a reference because it suggests it has a protective effect against many illnesses due to the high consumption of fruit and vegetables [16].

The dietary habits of Mexican adolescents exhibited significant deficiencies in terms of diversity. In addition, the consumption of vegetable foods should be promoted due to their association with beneficial effects on health, the prevention of non-communicable diseases, which are very common in the Mexican population, and if these healthy habits can be established from an early age, it is likely to prevent the development of pathologies in later stages of life.

## 5. Conclusions

This study showed that the diet of Mexican adolescents was characterized with little variability in the foods consumed: 28.20% of the population had a good consumption of milk (1 to 5 serving/day), 16.50% of sugars, and 16% of cereals, while only less than 15% had a good consumption of source animal foods, fruits and vegetables, and oils and fats. A variable diet contributes to providing the main nutrients (proteins, carbohydrates, and lipids); these foods also provide other compounds, such as antioxidants, fiber, vitamins, minerals, etc., that achieve health benefits. Therefore, it is necessary to implement nutritional strategies in this population to increase the frequency and variety of foods, such as the provision of healthy meals, education sessions, physical activity, and/or family intervention focused on modifying the school environment in this Mexican population.

## Figures and Tables

**Figure 1 pediatrrep-17-00078-f001:**
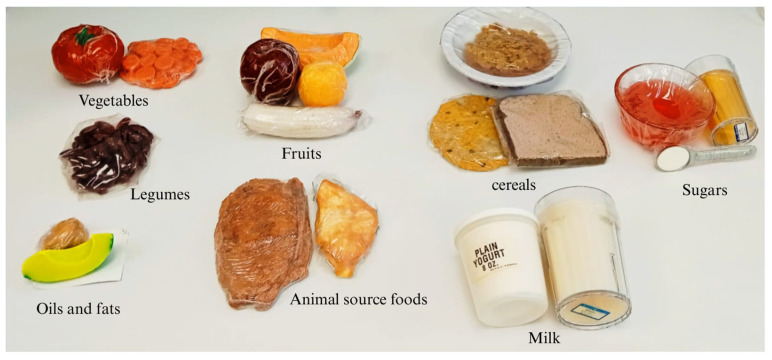
Food models and home-made portion sizes.

**Figure 2 pediatrrep-17-00078-f002:**
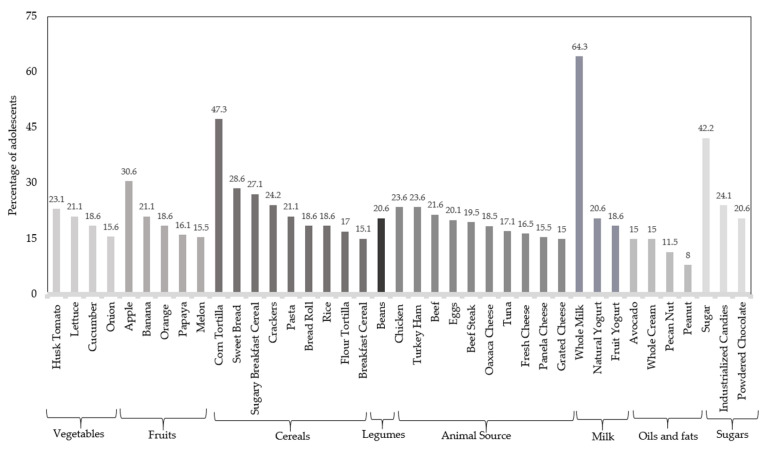
Percentage of food consumption in a population of adolescents (1–5 servings/day).

**Figure 3 pediatrrep-17-00078-f003:**
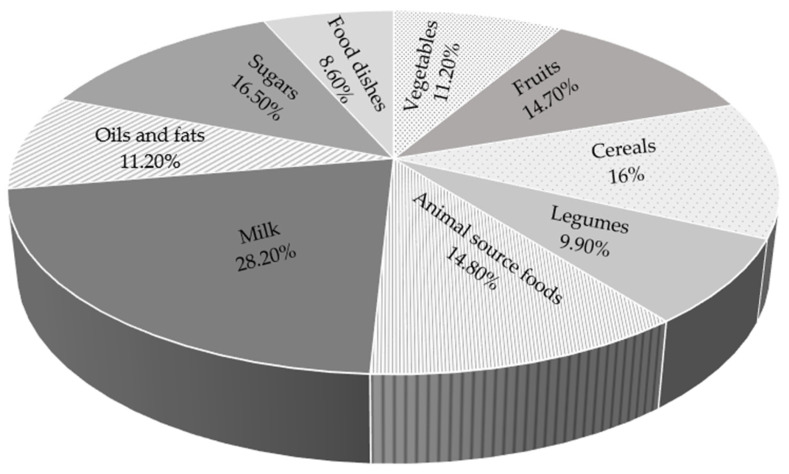
Food group consumption among Mexican adolescents.

**Table 1 pediatrrep-17-00078-t001:** Daily intake of the most frequently consumed foods of adolescents.

Food	G Fresh Matter of Edible Portion/Day/Person *	Equivalent Foods **	Number of Total Recommended Servings ***
Vegetables			
Cucumber	62.52 ± 94.01	0.60	
Husk Tomato	75.36 ± 122.04	0.88	
Lettuce	35.12 ± 47.18	0.26	
Onion	40.28 ± 66.37	0.69	
Tomato	53.65 ± 84.01	0.47	
Total	269.93	2.90	4
Fruits			
Apple	118.75 ± 163.66	1.12	
Banana	74.09 ± 114.75	1.37	
Melon	75.26 ±126.18	0.42	
Orange	46.89 ± 76.59	0.31	
Papaya	67.60 ± 124.80	0.48	
Total	382.59	3.70	3
Cereals			
Bread Roll	27.71 ± 41.24	1.38	
Breakfast Cereal	5.11 ± 9.64	0.27	
Corn Tortilla	41.73 ±44.86	1.39	
Crackers	5.29 ± 10.48	0.33	
Flour Tortilla	21.01 ± 35.36	0.75	
Pasta	80.60 ± 105.81	1.34	
Rice	38.64 ± 48.79	0.82	
Sugary Breakfast Cereal	8.48 ± 11.67	0.65	
Sweet Bread	34.60 ± 48.20	1.65	
Total	263.17	8.58	10
Legumes			
Beans	73.17 ± 96.79	0.85	2
Oils and fats			
Avocado	19.26± 32.00	0.62	
Japanese Peanut	7.43 ± 15.09	0.53	
Natural Peanut	7.43 ± 15.09	0.62	
Pecan Nut	2.35 ± 4.57	0.26	
Soy Oil	27.78	5.56	
Total	64.25	7.59	5
Sugars			
Powdered Chocolate	2.88 ± 4.78	0.29	3

Recommendations taken from “Guías alimentarias y de actividad física en contexto de sobrepeso y obesidad en población mexicana” [9]. * the grams of fresh matter of the edible portion/day/person indicate the weight on a fresh basis to quantify the intake. ** food equivalents corresponded with a portion or ration of food whose nutritional contribution is similar to those of the same group in quality and quantity, allowing them to be interchangeable with each other. *** recommended food serving is the amount of food from a specific food group, advice for consumption, and typically based on dietary guidelines or recommendations.

## Data Availability

No new data were created or analyzed in this study. Data sharing is not applicable to this article.

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
