# Peer review of "Pilot Study on the Evaluation of the Diet of a Mexican Population of Adolescents"

_pediatrrep, 2025, doi:10.3390/pediatric17040078_

Round 1
Reviewer 1 Report
Comments and Suggestions for Authors
Dear authors,
The chosen topic is a topical one and can be a starting point for developing of specific
interventions to improve the diet of adolescents in the region, implicitly to protect the health
of the future generation of adults.
For your study to be published, a few changes are needed:
1. The title should contain the phrase “pilot study” because the methodology is suitable
for this type of study, and the introduction, discussion and conclusions sections should be
limited to the evaluation of the diet and not to the evaluation of the intake of bioactive
compounds (antioxidants).
2. The aim of the study should be clearly stated: did you want to evaluate the overall diet
or to evaluate bioactive compounds? You cannot make assumptions about the intake of
bioactive compounds without having targeted questions for foods that are good sources of
bioactive compounds.
It would be preferable to separate this research from the bioactive compounds assessment
because they are different things.
3. The abstract should be rewritten according to the requirements of the journal Pediatric
Reports: Context/Objectives: A few sentences to place the question addressed in a broader
context and to highlight the purpose of the study. Methods: A brief description of the main
methods or treatments applied. This may include any relevant pre-registration or specimen
information. Results: A brief summary of the main findings of the article. Conclusions: A final
comment summarizing the main conclusions or interpretations. Please formulate in the
Introduction section, more clearly stated, the purpose of your study and the hypotheses that
led to this research.
4. Regarding the results it would be preferable to avoid repetition and be consistent
throughout the manuscript. For example, you have said that only 14.7% and 11.2% of
adolescents consume fruits and vegetables, respectively (lines 13-15 and 147-149) and in the
discussion you stated that in your study the consumption of fruits and vegetables also
complies with the recommendations of the WHO and The Spanish Society of Community
Nutrition (lines 161-162). Were you referring to the whole sample or just those who declared
daily consumption?
5. The conclusions section should be reworded in accordance with the instructions for
manuscript preparation.
6. English language verification is required.
7. It is necessary to revise the reference list if you separate this research from bioactive
compounds assessment.
Author Response
Dear editor and reviewers, we appreciate your support in improving this manuscript, which will enhance what is presented here.
Dear authors,
Please see the attachment, pleace.

Reviewer 2 Report
Comments and Suggestions for Authors
General
The manuscript addresses a relevant and timely public health topic—dietary intake and food habits among Mexican adolescents. The study offers original data using a validated FFQ, with implications for nutritional policy and adolescent health strategies. However, the manuscript would benefit from language improvement, clearer presentation of results, and a few clarifications.
Strengths
- Relevance: The focus on adolescents' diet in a Mexican urban setting is of high public health significance.
- Use of validated tools: The use of a semi-quantitative FFQ validated for Mexican adolescents enhances the credibility of dietary assessments.
- Ethical approval: Ethical protocols were followed, including informed consent from participants and guardians.
Major points for revision
- Language and grammar
- The manuscript contains multiple grammatical and syntactical issues (e.g., "Currently there are few o studies", "Therefore, it could be a valid and reliable instrument"). It requires comprehensive English editing by a native speaker or professional editor.
- Clarity and Structure
- Some tables and descriptions (especially Tables 2 and 3) are overly detailed or unclear. Summarizing key findings with visual clarity (perhaps use of graphs or reorganized tables) would improve readability.
- Figures mentioned (e.g., Figures 1–3) are not included in the peer review document. These should be reviewed to assess clarity and relevance.
- Statistical analysis
- Although standard tests are used (t-tests, chi-square), more details on statistical significance thresholds, effect sizes, or confidence intervals would add robustness.
- Clarify whether intake values were energy-adjusted or not.
- Introduction
- While informative, the introduction could more clearly define the gap in current research.
- Additional context on the role of polyphenols and bioactive compounds in adolescence would enhance relevance.
- Methods
- Further detail is needed on how food equivalents were calculated and how daily intakes were estimated.
- Clarify statistical procedures, including significance thresholds and how non-normal data were handled.
- Ensure definitions of food groups (e.g., “dishes”) are consistent and well explained.
- Results
- The tabular format is dense and could benefit from visual summaries (e.g., bar charts or pie graphs).
- Several figures mentioned (e.g., Figures 1–3) were not included in the review copy and should be carefully checked.
- Make clearer distinctions between grams, equivalents, and recommended servings.
- Discussion
- The discussion references national and international guidelines appropriately but needs deeper engagement with potential socioeconomic, cultural, and urban factors influencing dietary patterns.
- Consider elaborating on why legumes and vegetables are consumed infrequently and connect this to broader food system trends or adolescent behaviors.
- The discussion lacks depth in interpreting why consumption patterns are low, especially for legumes, and how socioeconomic factors may play a role.
- The link between low bioactive compound intake and health outcomes could be expanded with more critical engagement with recent literature
- Conclusion
- The conclusion is aligned with the study findings but should better highlight specific policy or educational recommendations.
- The mention of future biomarker-based research is promising—consider suggesting concrete next steps.
Minor issues
- Typos and formatting: “Autonumus” should be “Autónoma”, “Departament” should be “Department”, and spacing issues are found throughout the text.
- References: Ensure all references are formatted per MDPI guidelines. Some URLs are repeated multiple times and clutter the reference list.
The manuscript contains multiple grammatical and syntactical issues (e.g., "Currently there are few o studies", "Therefore, it could be a valid and reliable instrument"). It requires comprehensive English editing by a native speaker or professional editor. eg Autonumus, Departament
Author Response
Dear editor and reviewers, we appreciate your support in improving this manuscript, which will enhance what is presented here.
Dear authors,
Please see the attachment.

Reviewer 3 Report
Comments and Suggestions for Authors
Manuscript focuses on the evaluation of the diet in a convenient sample of Mexican adolescents.
Study has been approved by Ethics Committee.
Inclusion and exclusion ciriteria are clear and enable participant selection.
Validated FFQ was used and food models were used to facilitate quantification.
Instead od placing FFQ through figures, it should be provided as a supplementary material. Alternatively it is possible to simply cite reference where FFQ was previousely published. This would save a lot of space without loosing and important information.
Informaton on the steps between FFQ and statistical analysis is missing. How did you analyse FFQ data. Did you use some software or calculation was performed manualy? Please provide details!
Line 135: please delete superscript on the equivalents or othervise explain it.
Footnote on the table 2 is not connected to the table content. Please provide connection (asterisk maybe!)
Table 3 footnote is also partial. Connection missing for the "2"
Table 3 - it would be more useful to see the percentage of adolescents whos inatke is in line with the recommendations for selected foods. 1-5 portions a day is too much for some of the listed foods and on the other side not enough for some others.
Author Response

(The authors gave the same response as above.)

Round 2
Reviewer 1 Report
Comments and Suggestions for Authors
Please find attached my remarks at this stage.

Author Response
Dear editor and reviewers, we appreciate your support and time in improving this manuscript, which will enhance this investigation. We present the letter point to point of the corrections of the reviewers.

Reviewer 2 Report
Comments and Suggestions for Authors
Thank you for your revisions. The manuscript has improved significantly in clarity, structure, and discussion. The visual summaries and clearer distinctions in the tables really improved the readability of the results. The expanded discussion on socioeconomic and cultural factors, as well as the role of bioactive compounds, adds valuable context.
The authors have made improvements in response to the initial feedback. The topic remains highly relevant, and the study contributes valuable data on adolescent dietary patterns in a Mexican urban context.
The manuscript now presents a clearer structure, with reorganised tables and added figures that enhance readability.
The introduction and discussion sections have been expanded to better contextualise the research gap and socioeconomic influences on dietary habits.
Methodological details have been clarified, including food equivalents and statistical procedures.
However i feel the following needs addressing further:
While the English language has improved, the manuscript still contains grammatical and syntactical issues that affect clarity. A final round of professional English editing is strongly recommended.
Clarification is still needed on whether dietary intake values were energy-adjusted.
I recommend acceptance with minor revisions, assuming comprehensive English language editing and clarification of the energy adjustment in dietary intake values.
Comments on the Quality of English Language
Thank you for your revisions.
I feel that the manuscript still contains several grammatical and syntactical issues that affect the fluency and clarity of the text. A final round of professional English editing is strongly recommended to ensure the language meets publication standards.
Once the language is polished, the manuscript will be suitable for publication.
Author Response

(The authors gave the same response as above.)

Reviewer 3 Report
Comments and Suggestions for Authors
Congrats on the improovements!
Now this is much better.
Author Response
Dear editor and reviewers, we appreciate your support and time in improving this manuscript, which will enhance this investigation. We present the letter point to point of the corrections of the reviewers.
Thank you very much for your comments.

Round 3
Reviewer 1 Report
Comments and Suggestions for Authors
Dear authors,
Congratulations for the effort to improve the article! I give you my agreement for the publication after resolving the comments in the attached document.

Author Response
Dear Reviewer,
Thank you very much for taking the time to review our manuscript and for your comments.
